# Biomarker Insights: Evaluation of Presepsin, Apelin, and Irisin Levels in Cutaneous Leishmaniasis

**DOI:** 10.3390/diagnostics14242869

**Published:** 2024-12-20

**Authors:** Revsa Evin Canpolat-Erkan, Recep Tekin, Bilal Sula

**Affiliations:** 1Department of Clinical Biochemistry, Faculty of Medicine, Dicle University, 21280 Diyarbakir, Turkey; drevinerkan@gmail.com; 2Department of Infectious Diseases and Clinical Microbiology, Faculty of Medicine, Dicle University, 21280 Diyarbakir, Turkey; 3Department of Dermtoology, Faculty of Medicine, Dicle University, 21280 Diyarbakir, Turkey; drbilalsula@gmail.com

**Keywords:** cutaneous leishmaniasis, presepsin, apelin, irisin

## Abstract

**Background/Objectives**: Cutaneous leishmaniasis (CL) is a skin disease caused by *Leishmania* parasites. Presepsin, irisin, and apelin are biomarkers that are involved in the inflammatory response. The aim of this study was to investigate the association between serum levels of specific biomarkers, such as presepsin, apelin, and irisin, and the clinical features, location, number, and size of lesions in patients with CL. **Methods**: This study is a single-centre, prospective cohort study involving a total of 30 patients with skin lesions compatible with CL and 30 healthy matched controls. Age, sex, type of skin lesion, location of skin lesion, number of skin lesions, and diameter of skin lesions were recorded. The levels of presepsin, irisin, and apelin measured in the blood samples of the patient group were analysed in comparison to those in the healthy control group. **Results**: The findings revealed that presepsin levels were significantly elevated in the patient group compared to the controls (*p* = 0.000). However, no statistically significant differences were observed between the groups for irisin and apelin levels (*p*-values 0.096 and 0.836, respectively). A negative correlation was identified between presepsin levels and the number of skin lesions, the diameter of the largest lesion, and the total diameter of the lesions (*p* = 0.000). **Conclusions**: It appears that measuring presepsin levels in patients with CL may be beneficial. Presepsin has the potential to serve as a prognostic marker in CL, offering significant benefits in guiding clinicians in assessing disease progression and response to treatment.

## 1. Introduction

Cutaneous leishmaniasis (CL) is a skin disease caused by *Leishmania* parasites [1]. The disease is characterised by the formation of long-lasting nodulo-ulcerative wounds on the skin that heal with atrophic cicatrix [2,3]. Its prevalence is significant, particularly in tropical and subtropical regions, affecting millions worldwide and posing substantial public health challenges. Despite its prevalence and public health impact, CL remains a neglected tropical disease, often overshadowed by other global health concerns. The occurrence of cutaneous leishmaniasis and the healing process are determined by the characteristics of the host, parasite, and vector. Despite advances in understanding its pathogenesis and transmission, effective management and control remain difficult due to drug resistance, vector ecology, and host immune responses [4,5,6]. Understanding the immunological and biochemical changes associated with CL is critical to advancing diagnostic and therapeutic approaches. In the quest for improved diagnosis and treatment of cutaneous leishmaniasis, the identification and evaluation of reliable biomarkers has become increasingly important [7,8]. Biomarkers are measurable indicators of a biological state or condition, and in the context of CL, they can help elucidate the host response to infection, guide treatment decisions, and improve prognostic assessments. Biomarkers can provide insight into the pathophysiology of the disease, guide therapeutic interventions, and help monitor treatment efficacy and disease progression [9]. By understanding the biochemical and molecular changes in infected individuals, researchers can develop targeted strategies for early diagnosis, personalised treatment plans, and potentially preventive measures against the disease. Recent studies have highlighted the importance of molecular biomarkers such as presepsin, irisin, and apelin in the progression of various diseases [10]. The objective of this study was to investigate the relationship between serum levels of specific biomarkers, such as presepsin, apelin, and irisin, and the clinical features, location, number, and size of lesions in patients suffering from cutaneous leishmaniasis.

## 2. Materials and Methods

This study is a single-centre, prospective cohort study. A total of 30 patients with skin lesions compatible with CL (smear-positive and/or pathology-positive and clinically compatible patients) and a healthy control group were included in this study. The province of Diyarbakir is located in the southeastern Anatolian region of Turkey. The patients’ diagnosis and treatment were supervised by medically qualified persons. Age, sex, type of skin lesion, location of skin lesion, number of skin lesions, diameter of skin lesions, and duration of skin lesion were recorded. The diagnosis of cutaneous leishmaniasis was made by cleaning the skin lesions of patients with nodulo-ulcerative lesions with alcohol and then scraping the material with a scalpel, taking the serosity in the ulcer with the help of a Pasteur pipette from the ulcerated lesions, spreading it on slides, staining it with Giemsa stain, and identifying the parasite in the lesions by direct smears in the laboratory.

Blood samples were taken from the brachial vein of individuals in the patient and control groups after a 12 h fast. For the control group, blood samples were also taken from 30 people with a body mass index (BMI) of less than 25, who had no chronic illnesses and were not taking any regular medication. Blood samples for presepsin, apelin, and irisin were collected in plain biochemistry tubes and centrifuged at 2000× *g* for 20 min, and the serum obtained was stored in Eppendorf at −80 °C until assayed. The levels of presepsin, apelin, and irisin in the serum were analysed by ELISA. After the sera were brought to room temperature and thawed on the day of the study, serum presepsin, apelin, and irisin were measured using the commercially available enzyme-linked immunosorbent assay kit (ELISA) method according to the kit contents [6,10]. The results of the study were compared and evaluated using various statistical methods.

Patients with non-ulcerated skin lesions (plaques, nodules, or verrucous and papular lesions), autoimmune disease, a history of other malignant disease, smear-negative patients, and pregnant women were excluded from the study.

### 2.1. Measurement of Serum Presepsin

Serum presepsin levels were determined using a commercial quantitative enzyme-linked immune sorbent assay (ELISA) technique (Sunred Biotechnology Company, Shanghai, China), in accordance with the manufacturer’s instructions. Serum presepsin levels were expressed in pg/mL.

### 2.2. Measurement of Serum Apelin

Serum apelin levels were determined using a commercially available quantitative enzyme-linked immune sorbent assay (ELISA) technique (Sunred Biotechnology Company, Shanghai, China), in accordance with the manufacturer’s instructions. Serum apelin levels were expressed in ng/mL.

### 2.3. Measurement of Serum Irisin

Serum irisin levels were determined by a commercial quantitative enzyme-linked immune sorbent assay (ELISA) technique (Sunred Biotechnology Company, Shanghai, China), in accordance with the manufacturer’s instructions. Serum irisin levels were expressed in ng/mL.

### 2.4. Statistical Analysis

The SPSS 22.0 (SPSS, IBM Corp., Armonk, NY, USA) for Windows package was used for the statistical analysis of the data obtained in the study. The Kolmogorov–Smirnov test was used to confirm that the data in both groups were within the ranges of normal distribution. A non-parametric test was used for variables outside the normal distribution. The Mann–Whitney U test or Student’s *t*-test was used to compare continuous parameters between reciprocal groups. The Chi-square test was used for the analysis of categorical parameters. The Spearman or Pearson test was used to analyse correlations, as appropriate. Statistical significance was based on a value of *p* < 0.05 with a 95% confidence interval.

## 3. Results

A total of 30 patients with skin lesions compatible with CL were included in this study. All patients live in or had visited areas in different CL endemic regions. Based on the laboratory and specimens, samples were taken from all patients and all of the participants tested positive. All collected samples were stained with Giemsa stain.

The sex and age status of the subjects were matched between the groups. Among the 30 patients with CL, 16 (53%) were male and 14 (47%) were female, with a mean age of 46 ± 17.2 years. In total, 15 patients (50%) had one lesion, 5 patients (17%) had two lesions, 1 patients (3%) had three lesions, and 9 patients (30%) had four or more lesions. Seventeen patients (57%) had lesions localised in the head and neck, seven patients (23%) had lesions localised in both the superior and inferior extremities, four patients (13%) had lesions localised in the inferior extremities, and two patients (7%) had lesions localised in the superior extremities (Table 1). Most of the lesions were located on the head and neck, the most affected areas. A few patients had multiple lesions on the hand, thigh, or neck. The duration of the disease ranged from 1 month to 36 months.

The levels of presepsin, irisin, and apelin present in the blood samples obtained from the patient cohort were compared with those observed in the blood of the healthy control group. The presepsin levels were found to be significantly elevated in the patient group in comparison to the control group (*p* = 0.000) (Figure 1).

There was no significant difference between the two groups for irisin and apelin (*p*-values 0.096 and 0.836, respectively) (Table 2). In order to evaluate the potential correlation between presepsin levels and the severity of CL, a Spearman correlation was performed on all patients. A negative correlation was observed between presepsin levels and the number of skin lesions (Figure 2), the diameter of the largest lesion, and the total diameter of the lesions (*p* = 0.000) (Figure 3). However, no significant correlation was identified between other parameters.

## 4. Discussion

The evaluation of biomarkers in parasitic diseases such as cutaneous leishmaniasis provides invaluable information on disease mechanisms, potential diagnostic tools, and therapeutic targets. Although many markers have been studied that influence the mechanisms of disease development, disease severity, and treatment response in CL, there are still many areas that remain to be investigated [11]. The host immune response to leishmaniasis varies depending on the type and virulence of the parasite and the immune response of the host. Because *Leishmania* protozoans are intracellular parasites, they hide within macrophages and are protected from the B cell antibody response [1]. The host’s initial immune response to leishmaniasis is established by innate immunity, and the response continues with acquired immunity. While macrophages, neutrophils, natural killer cells, and dendritic cells are involved in the innate immune response, T cell-mediated immunity and released cytokines are responsible for the actual resistance to leishmaniasis [12,13]. Proinflammatory cytokines such as TNF-α, IFN-β, IL-1, IL-6, IL-8, IL-12, and IL-17 are secreted first in CL. Then, anti-inflammatory cytokines such as IL-5, IL-4, IL-10, IL-13, and TGF-β are secreted [13]. Koçyiğit et al. found high levels of IL-1β, IL-8, IL-6, and TNF-α in patients with CL [14]. Although recent studies have focused on IL and TNF-alpha, there are very few studies on novel molecules. In recent years, attention has been focused on specific novel biomarkers such as presepsin, apelin, and irisin due to their potential role in immune response modulation and metabolic regulation in the context of infectious diseases [13,15]. The objective of this study was to examine the potential correlation between serum levels of specific biomarkers, including presepsin, apelin, and irisin, which have not been previously investigated in the presence of cutaneous leishmaniasis, and the clinical features, location, number, and size of lesions in patients with cutaneous leishmaniasis.

Presepsin is a protein fragment produced by the proteolytic degradation of the CD14 receptor on the surface of macrophages and other immune cells in inflammatory conditions such as infection and sepsis. This protein fragment occurs as a response of the body to infection [16,17,18]. In a review by Formenti et al., it was emphasised that presepsin is found at high levels during sepsis and inflammatory conditions and may help to determine the prognosis of patients. The authors also emphasised that the use of presepsin as a rapid and reliable marker may provide important advantages to healthcare professionals in decision-making processes in critical patient management [14]. Masson et al. and Zhang et al. showed that presepsin is a useful marker for monitoring inflammatory processes, especially in bacterial infections and sepsis [19,20]. Shiota et al. demonstrated that the usefulness of presepsin for the diagnosis of skin wound infection in haemodialysis patients was comparable to that of sepsis and superior to conventional biomarkers of infection. They found that the cut-off value for diagnosing skin infection was close to that used to predict the prognosis of foot gangrene [21]. Furthermore, Ha et al. found a significant increase in serum presepsin levels in patients with diabetic foot ulcers. In this study of patients with diabetic foot ulcers, they stated that presepsin levels are a strong marker for predicting the severity of infection and that presepsin is an effective biomarker for monitoring the degree of infection and response to treatment [22]. In our study, similarly to the study by Masson et al. [19], Zhang et al. [20], and Ha et al. [22], we found that presepsin levels (1358.12 pg/mL) were higher in patients with CL compared to the control group (466.6; *p* = 0.000). We also observed a negative correlation between presepsin levels and the number and size of lesions. This is a first in the literature. Elevated presepsin levels in CL patients suggest that this biomarker reflects the inflammatory response and may be important in the pathogenesis of leishmaniasis. In this study, presepsin levels were negatively correlated with lesion number, largest lesion diameter and total lesion diameter, suggesting that CL severity may be associated with presepsin levels. It is possible that lesion size and number do not increase the inflammatory burden of the disease differently from other diseases and that this inflammatory response is reflected in presepsin levels. Presepsin can also be considered as a potential biomarker to monitor the degree of reduction in infection and inflammation during the treatment process. We believe that the use of presepsin in parasitic infections such as CL will attract attention as a new field with this study. However, further research is needed.

Irisin is a peptide secreted from muscle tissue in response to exercise [6]. It plays a role in energy homeostasis and is associated with a number of biological processes that affect inflammatory processes [23]. However, recent studies have shown that it increases anti-inflammatory cytokines and decreases proinflammatory cytokines. Irisin causes a decrease in the migration and infiltration of macrophages as well as a decrease in the acute phase inflammatory response [24]. Irisin levels may vary in inflammatory states, but information on the level of this biomarker in inflammation caused by parasitic infections is limited. Ambrogio et al. investigated serum irisin levels in patients with chronic plaque psoriasis and determined that irisin levels were significantly higher in patients with chronic plaque psoriasis compared to control groups [25]. In their study, which included 171 subjects (115 acne patients of varying severity and 56 healthy controls), Tang et al. observed a significant reduction in serum irisin levels among patients with acne vulgaris [26]. The present study did not identify a significant difference in irisin levels between the study group and the control group. Furthermore, no significant increase in irisin levels was observed in CL patients, which suggests that irisin is not directly related to CL. It is recommended that future studies elucidate the effects of irisin levels in CL and its place in clinical applications.

Apelin plays an important role in energy metabolism, angiogenesis, and inflammatory response and has been shown to have important effects on infection and inflammation [27,28]. Apelin, which can bind to angiotensin type 1 receptor-associated proteins defined as orphan GPCRs, is a member of the family of peptides called the apelinergic system. Apelin has been shown to be localised in vascular endothelial cells and adipose tissue, brain, lung, heart, gastrointestinal system, uterus, ovary, and placenta. Apelin has cardiorenal protective, antihypertensive, positive inotropic, and anti-inflammatory effects and has been shown to regulate energy metabolism, apoptosis, and oxidative stress. Apelin reduces IL-1β and induces the release of IL-6 and IL-8 [29]. In a study by Sandal et al., research on apelin biomarkers highlighted the potential role of this molecule in inflammation and various diseases. It is thought that apelin may have regulatory effects in infection and inflammatory processes. Therefore, the determination of apelin levels can be considered as a useful biomarker in the diagnosis and treatment monitoring of inflammatory diseases [30]. In an experimental study, Yamazaki et al. observed an increase in apelin levels in mice subjected to cutaneous ischaemia [31]. In our study, there was no significant difference in apelin levels compared to the control group. Further research is required to explain the precise mechanisms by which apelin influences the immune response in cutaneous leishmaniasis and to explore its prospective role in targeted therapies.

The main limitations of our study are the small sample size and the fact that it was conducted in a single centre. In studies to be conducted in different geographical regions and with larger sample sizes, it may be possible to evaluate biomarkers as more general biomarkers in the diagnosis and follow-up of CL.

## 5. Conclusions

The serum presepsin levels were higher in CL patients compared to the healthy population. Presepsin may be an important biomarker for the correlation of lesion number and diameter with prognosis in CL. Presepsin can be used as a potential prognostic marker in CL and may be of great benefit in guiding clinicians in assessing disease progression and response to treatment. Further longitudinal and prospective studies are needed to clarify the pathophysiological role of presepsin levels in disease severity and lesion number.

## Figures and Tables

**Figure 1 diagnostics-14-02869-f001:**
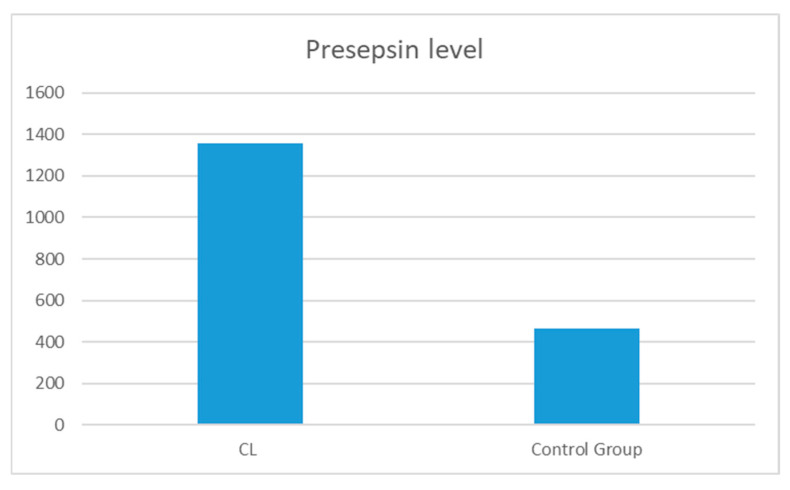
Levels of presepsin in the CL group and the control group.

**Figure 2 diagnostics-14-02869-f002:**
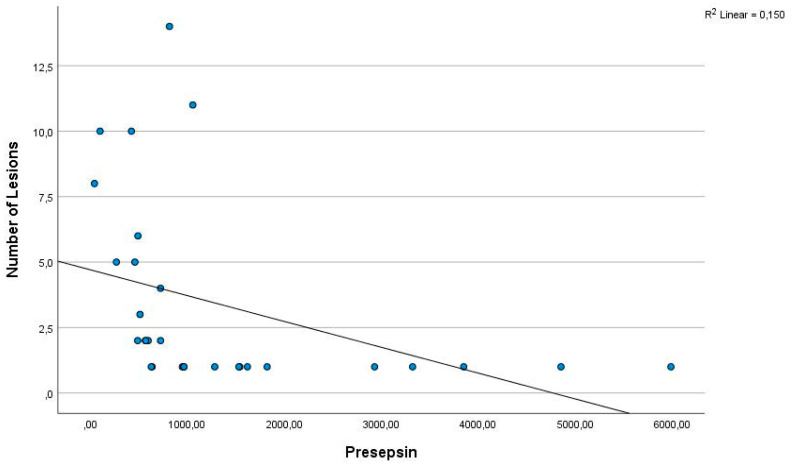
Linear regression curves between the presepsin levels and the number of skin lesions, which showed significant correlations in Pearson’s correlation analysis. The presepsin levels showed a negative correlation with the number of skin lesions (r = −0.728; p = 0.000).

**Figure 3 diagnostics-14-02869-f003:**
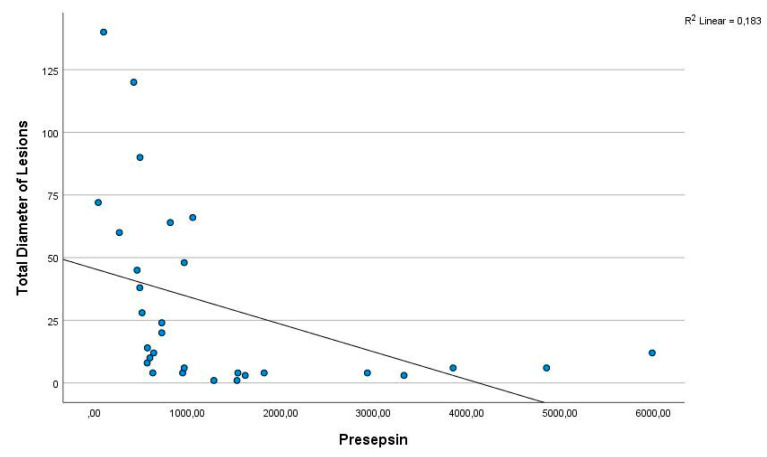
The presepsin levels showed a negative correlation with the total diameter of the lesions (r = −0.695; p = 0.000).

**Table 1 diagnostics-14-02869-t001:** Clinical features and characteristics of the lesions.

	Total *n* = 30 (%)
Age (year)	
	46.1 ± 17.2
Gender	
Female	14 (43)
Male	16 (57)
Time of Lesion (month)	
<6 months	22 (73)
6–12 months	6 (20)
>12 months	2 (7)
Location of Lesion	
Head and neck	17 (57)
Multiple location	7 (23)
Lower limbs	4 (14)
Upper limbs	2 (7)
Number of Lesions	
Single	15 (50)
Two	5 (17)
Multiple >3	10 (33)
Size of Lesion	
<15 mm	17 (57)
15–30 mm	3 (10)
30–45 mm	2 (7)
>45 mm	8 (26)
Type of Lesion	
Dry	18 (60)
Wet	12 (40)

**Table 2 diagnostics-14-02869-t002:** Presepsin, irisin, and apelin levels in the study group and the control group (mean ± standard deviation).

	Cutaneous Leishmania Patients (*n* = 30)	Controls (*n* = 30)	*p*
Presepsin (pg/mL)	1358.12 ± 1431.7	466.6 ± 531.1	0.000
Irisin (ng/mL)	7.1 ± 9.1	5.5 ± 2.6	0.096
Apelin (ng/mL)	18.2 ± 2.65	17.8 ± 9.8	0.836

## Data Availability

Dataset available on request from the authors. The raw data supporting the conclusions of this article will be made available by the authors on request.

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
