# Peer review of "Biomarker Insights: Evaluation of Presepsin, Apelin, and Irisin Levels in Cutaneous Leishmaniasis"

_diagnostics, 2024, doi:10.3390/diagnostics14242869_

Round 1
Reviewer 1 Report
Comments and Suggestions for Authors
Abstract (line 12-28):
Background: Give a short description of the studied markers.
Methods: Add the study area in and the number of samples.
Introduction (line 32-57):
Scientific names (like Leishmania) should be italic in the whole manuscript
There is no clear explanation of the importance of the studied markers (presepsin, apelin and irisin) in the introduction. Authors should highlight the role and importance of these markers in the parasitic disease, particularly in leishmaniasis.
Materials and methods:
Line 59-69: how did the samples were diagnosed as cutaneous leishmaniasis?
What is the species of Leishmania?
What hospitals/medical centers/ geographical area were the samples been collected?
Add suitable references for lab work.
Line 70-80: add suitable references
Add the ethical approval for human sample collection (number & date)
Line 84-98: add the selectivity of each kit
Results:
Line 111: where is the endemic area located? Altitude? Year/month of sample collection?
The authors did not mention the results of the gender (male and female) + age grouping regarding the studied markers.
Discussion:
So many general information, discussion needs briefing with focus on the target results.
Conclusion:
No need to RE write the results. Overrated conclusion, needs briefing and to be compatible with the small size of samples.
Comments on the Quality of English Language
The manuscript needs to be proofread by English native speaker.
Author Response
We have corrected our article in accordance with reviewer recommendations.
Reviewer 1:
- Abstract: Background: Give a short description of the studied markers.
Response: We added this sentence ‘Presepsin, irisin and apelin are biomarkers that are involved in the inflammatory response’.
- Materials and methods: Add the study area in and the number of samples
Response: We changed this sentence ‘This study is a single-center, prospective cohort study a total of 30 patients with skin lesions compatible with CL were included in this study’ as ‘This study is a single-center, prospective cohort study involving a total of 30 patients with skin lesions compatible with CL and 30 healthy matched controls’
- Introduction (line 32-57):Scientific names (like Leishmania) should be italic in the whole manuscript.
Response: We changed Leishmania as ‘Leishmania’ in the whole manuscript.
- Introduction (line 32-57): There is no clear explanation of the importance of the studied markers (presepsin, apelin and irisin) in the introduction. Authors should highlight the role and importance of these markers in the parasitic disease, particularly in leishmaniasis.
Response: Since the biomarkers used in the study have not been previously studied in parasitic infections, we could not provide detailed literature information.We gave information with these sentences in the introduction; ‘Biomarkers are measurable indicators of a biological state or condition, and in the context of CL they can help elucidate the host response to infection, guide treatment decisions and improve prognostic assessments. Biomarkers can provide insight into the pathophysiol-ogy of the disease, guide therapeutic interventions, and help monitor treatment efficacy and disease progression. By understanding the biochemical and molecular changes in infected individuals, researchers can develop targeted strategies for early diagnosis, per-sonalised treatment plans and potentially preventive measures against the disease. Recent studies have highlighted the importance of molecular biomarkers such as presepsin, irisin and apelin in the progression of various diseases’.
- Materials and methods: Line 59-69: how did the samples were diagnosed as cutaneous leishmaniasis?
Response: We write this sentence ‘skin lesions compatible with CL (smear positive and/or pathology positive and clinically compatible patients) and healthy control group were included in this study. The diagnosis of cutaneous leishmaniasis was made by cleaning the skin lesions of the patients with nodulo-ulcerative lesions with al-cohol and then scraping the material with a scalpel and taking the serosity in the ulcer with the help of a pasteur pipette from the ulcerated lesions, spreading it on slides and staining it with Giemsa stain and showing the parasite in the lesions by direct smears in the laboratory’.
- What is the species of Leishmania?
Response: We do not make Leishmania species.
- What hospitals/medical centers/ geographical area were the samples been collected?
Response: We added this sentence ‘The province of Diyarbakir is located in the southeastern Anatolian region of Turkey.
- Add suitable references for lab work.
Response: We added this references ‘Bahrami, F., Harandi, A. M., Rafati, S. Biomarkers of Cutaneous Leishmaniasis. Frontiers in cellular and infection microbiolo-gy. 2018, 8, 222. and ‘Canpolat Erkan, RE., Tekin, R. Investigation of new inflammatory biomarkers in patients with brucella. PLoS One 2024, 19, e0297550’.
- Line 70-80: add suitable references
Response: We added this references ‘Bahrami, F., Harandi, A. M., Rafati, S. Biomarkers of Cutaneous Leishmaniasis. Frontiers in cellular and infection microbiolo-gy. 2018, 8, 222.
- Add the ethical approval for human sample collection (number & date)
Response: We have included our ethical statement in the Institutional Review Board Statement section of our article.
Institutional Review Board Statement: The study was conducted in accordance with the Declara-tion of Helsinki and approved by the Health Sciences University Diyarbakır Gazi YaÅŸargil Training and Research Hospital Ethics Committee (protocol code 559; date of approval—11 sep-tember 2020).
- LIne 84-98: add the selectivity of each kit
Response: We have included the selectivity of each kit in the Methods section of our article.
- Results: Line 111: where Is the endemic area located? Altitude? Year/month of sample collection?
Response: We added this sentence ‘The province of Diyarbakir is located in the southeastern Anatolian region of Turkey’.
- The authors did not mention the results of the gender (male and female) + age grouping regarding the studied markers.
Response: As we did not find a correlation between the markers analysed and age and sex, we did not feel the need to write this sentence in detail in the result section; ‘However, no significant correlation was identified between other parameters’.
- Discussion: So many general Information, discussion needs briefing with focus on the target results.
Response: Since the biomarkers used in the study have not been studied in parasite infections before, we could not find detailed literature information and we thought that it would be useful to provide detailed information since this is the first study in this field.
- Conclusion: No need to RE write the results. Overrated conclusion, needs briefing and to be compatible with the small size of samples.
Response: We removed this sentence ‘Presepsin may be an important biomarker for the correlation of lesion number and diameter with prognosis in CL’ in the conclusion section of our article.
Reviewer 2 Report
Comments and Suggestions for Authors
The manuscript “Biomarker Insights: Evaluation Presepsin, Apelin and Irisin Levels in Cutaneous Leishmaniasis” submitted to the journal Diagnostics is deal with the investigation the association between serum levels of specific biomarkers, such as presepsin, apelin and irisin, and the clinical features, location, number and size of lesions in patients with cutaneous leishmaniasis. The topic discussed is rather interesting.
I would like to make a few comments:
1. In the Materials and Methods section, please indicate the manufacturers of all reagents, kits as well as the names and manufacturers of all equipment used.
2. Lines 70-72: “Blood samples were taken from the brachial vein of individuals in the patient and control groups after a 12-hour fast. Blood samples were also taken from 30 people with a body mass index (BMI) of less than 25, who had no chronic illnesses and were not taking any regular medication.”
Comment 1: How many groups were examined? According to the text there were 3 (1-with CL; 2-control; 3- “people with a body mass index (BMI) of less than 25”.
Comment 2: The text does not mention body mass index anywhere else, including in the group with SL. Is information on BMI necessary for the control group?
3. Line 113: “All collected samples were stained with gram stain”
Comment: Was it really Gram staining? In Line 68 it is written “staining it with Giemsa stain”.
4. In Figure 1, the data scatter and the significance of differences between columns should be indicated.
5. Line 180: There is a mistake; IL-6 is proinflammatory cytokine, not anti-inflammatory.
6. Lines 179-181: “Proinflammatory cytokines such as TNF-α, IFN-β, IL-1, IL-8, IL-12, IL-17 are secreted first in CL. Then, anti-inflammatory cytokines such as IL-5, IL-6, IL-4, IL-10, IL-13 and TGF-β are secreted.”
Comment: The reference is needed.
7. Line 196: “the use of resepsin”.
Comment: There is a mistake, correct is presepsin.
8. Line 248: “Apelin reduces IL-1β and induces the release of IL-6 and IL-8 [28].”
Comment: there is no information about IL-1β, IL-6 and IL-8 in the reference [28].
9. Please explain what is the benefit of finding a new marker in leishmaniasis? Can the authors explain the possible mechanism of CD14 shedding and presepsin increase in leishmaniasis.
Author Response
Response to reviewer
We have corrected our article in accordance with reviewer recommendations.
Reviewer 2:
- In the Materials and Methods section, please indicate the manufacturers of all reagents, kits as well as the names and manufacturers of all equipment used.
Response: We indicated all of the kits names and manufacturers in the Methods section of our article. Like this kit;
‘Serum presepsin levels were determined using a commercial quantitative enzyme-linked immune sorbent assay (ELISA) technique (Sunred Biotechnology Company, Shanghai, China), in accordance with the manufacturer's instructions. Serum presepsin levels were expressed in pg/mL.’
- LInes 70-72: “Blood samples were taken from the brachIal veIn of IndIvIduals In the patIent and control groups after a 12-hour fast. Blood samples were also taken from 30 people wIth a body mass Index (BMI) of less than 25, who had no chronIc Illnesses and were not takIng any regular medication.”
Comment 1: How many groups were examined? According to the text there were 3 (1-wIth CL; 2-control; 3- “people with a body mass Index (BMI) of less than 25”.
Comment 2: The text does not mention body mass index anywhere else, Including in the group with SL. Is Information on BMI necessary for the control group?
Response: There was a misunderstanding caused by a grammatical error in this sentence. there are two groups in the study, patient and control group. necessary corrections were made. We changed this sentence ‘Blood samples were taken from the brachial vein of individuals in the patient and control groups after a 12-hour fast. Blood samples were also taken from 30 people with a body mass index (BMI) of less than 25, who had no chronic illnesses and were not taking any regular medication’’ as ‘Blood samples were taken from the brachial vein of individuals in the patient and control group after a 12-hour fast. For the control group, blood samples were also taken from 30 people with a body mass index (BMI) of less than 25, who had no chronic illnesses and were not taking any regular medication’
- LIne 113: “All collected samples were staIned with gram stain”
Comment: Was It really Gram staInIng? In LIne 68 It Is wrItten “staInIng It wIth GIemsa staIn”.
Response: We correct this sentence ‘Gram staining’ as Giemsa stain”.
- In Figure 1, the data scatter and the significance of differences between columns should be indicated.
Response: We have indicated be this sentence ‘The levels of presepsin, irisin and apelin present in the blood samples obtained from the patient cohort were compared with those observed in the blood of the healthy control group. The presepsin levels were found to be significantly elevated in the patient group in comparison to the control group (p=0.000) (Figure 1)’ in the result section of our article.
- LIne 180: There is a mistake; IL-6 is proinflammatory cytokine, not anti-inflammatory.
Response: We changed as IL-6 is proinflammatory cytokine. ‘Proinflammatory cytokines such as TNF-α, IFN-β, IL-1, IL-6, IL-8, IL-12, IL-17 are secreted first in CL. ‘
- LInes 179-181: “ProInflammatory cytokInes such as TNF-α, IFN-β, IL-1, IL-8, IL-12, IL-17 are secreted fIrst In CL. Then, antI-Inflammatory cytokInes such as IL-5, IL-6, IL-4, IL-10, IL-13 and TGF-β are secreted.”
Comment: The reference is needed.
Response: We added this references ‘Divenuto, F., Pavia, G., Marascio, N., Barreca, G. S., Quirino, A., Matera, G. Role of Treg, Breg and other cytokine sets in host protection and immunopathology during human leishmaniasis: Are they potential valuable markers in clinical settings and vaccine evaluation?. Acta tropica 2023, 240, 106849.’
- LIne 196: “the use of resepsin”.
Comment: There is a mistake, correct is presepsin.
Response: We changed resepsin as ‘presepsin’
- LIne 248: “ApelIn reduces IL-1β and Induces the release of IL-6 and IL-8 [28].”
Comment: there Is no InformatIon about IL-1β, IL-6 and IL-8 In the reference [28].
Response: We corrected references. As ‘Wang, X., Zhang, L., Li, P., Zheng, Y., Yang, Y., Ji, S. Apelin/APJ system in inflammation. International immunopharmacology 2022, 109, 108822.’
- Please explain what is the benefit of finding a new marker in leishmaniasis? Can the authors explain the possIble mechanism of CD14 shedding and presepsin increase in leishmaniasis.
Response: In our study, we think that the high level of presepsin in CL is due to the fact that it is a molecule involved in the inflammatory response, as in other studies, but we could not show this in our study because it is necessary to show the mechanisms by which it increases, especially with molecular studies. we aim to find explanatory mechanisms in our future studies. in the discussion section of our article, we tried to explain the high level of presepsin with other literature information.